# Preterm disparities between foreign and Swedish born mothers depend on the method used to estimate gestational age. A Swedish population-based register study

Sol P. Juárez[1,2☯]*, Marcelo L. Urquia[3,4☯], Eleonora Mussino[5], Can Liu[1,2,6], Yao Qiao[4¤], Anders Hjern[1,2,6]

1 Centre for Health Equity Studies (CHESS), Stockholm University/Karolinska Institutet, Stockholm, Sweden, 2 Department of Public Health Sciences, Stockholm University, Stockholm, Sweden, 3 Department of Community Health Sciences, Rady Faculty of Health Sciences, University of Manitoba, Winnipeg, Canada, 4 Dalla Lana School of Public Health, University of Toronto, Toronto, Canada, 5 Stockholm University Demography Unit (SUDA), Stockholm University, Stockholm, Sweden, 6 Clinical Epidemiology Division/ Department of Medicine, Karolinska Institutet, Stockholm, Sweden

☯ These authors contributed equally to this work.
¤ Current address: Johns Hopkins Bloomberg School of Public Health, Johns Hopkins University, Baltimore, Maryland, United States of America
* sol.juarez@su.se

**Data Availability Statement:** Data cannot be shared publicly because of confidentiality. In order to access Swedish register data, researchers must

## Abstract

This study aims to examine whether disparities in gestational age outcomes between foreign and Swedish-born mothers are contingent on the measure used to estimate gestational age and, if so, to identify which maternal factors are associated with the discrepancy. Using population register data, we studied all singleton live births in Sweden from 1992–2012 (n = 1,317,265). Multinomial logistic regression was performed to compare gestational age outcomes classified into very (<32 weeks) and late preterm (32–36 weeks), term and post-term derived from the last menstrual period (LMP) and ultrasound estimates in foreign- and Swedish-born women. Compared to Swedish-born women, foreign-born women had similar odds of very preterm birth (OR: 0.98 [95% CI: 0.98, 1.01]) and lower odds of moderately preterm birth (OR: 0.95 [95% CI: 0.92, 0.98]) based on ultrasound, while higher risks based on LMP (respectively, OR: 1.10 [95% CI: 1.07, 1.14] and 1.09 [95% CI: 1.06, 1.13]). Conclusions on disparities in gestational age-related outcomes by mother's country of origin depend on the method used to estimate gestational age. Except for very preterm, foreign-born women had a health advantage when gestational age is based on ultrasound, but a health disadvantage when based on LMP. Studies assessing disparities in very preterm birth by migration status are not affected by the estimation method but caution should be taken when interpreting disparities in moderately preterm and preterm birth rates.

contact the agencies responsible for data protection, conditional on ethical vetting (contacts below). The National Board of Health and Welfare (Socialstyrelsen) - Postal address: Socialstyrelsen SE-106 30, Stockholm, Sweden; Visiting address: Rålambsvägen 3, Stockholm; Phone: +46 (0)75 247 30 00; Fax: +46 (0)75 247 32 52; E-mail: socialstyrelsen@socialstyrelsen.se. Statistics Sweden (SCB) - Postal address: Statistics Sweden, Solna strandväg 86, SE-171 54 Solna, Sweden; Visiting address: Solna strandväg 86; Phone: +46 (0) 10 479 40 00; E-mail: scb@scb.se. Swedish Ethical Review Authority (Etikprövningsmyndigheten) - Postal address: Box 2110, SE-750 02 Uppsala, Sweden; E-mail: registrator@etikprovning.se; Phone: +46 (0)10 475 08 00.

**Funding:** Sol P. Juárez and Eleonora Mussino were supported by the Strategic Research Council of the Academy of Finland (TITA, Grant # 293103) and the Swedish Research Council (VR# #2018-01825); Sol P. Juárez and Anders Hjern were supported by the Swedish Research Council for Health, Working Life and Welfare (FORTE grant #2012-01190 and #2016-07128). Marcelo L. Urquia holds a Canada Research Chair in Applied Population Health. The funders had no role in study design, data collection and analysis, decision to publish, or preparation of the manuscript.

**Competing interests:** The authors have declared that no competing interests exist.

## Introduction

Preterm birth is an important indicator for perinatal surveillance; it accounts for more than 80% of neonatal deaths [1] and is a risk factor for a number of adverse outcomes across the life course [2, 3]. Large differences in preterm rates exist between immigrant populations residing in high-income countries [4, 5]. Although heterogeneity has been linked to study designs and definitions of migrants and reference groups [5], the extent to which preterm differences between groups and contexts is affected by varying methods of estimation of the gestational age is still unknown.

Ultrasound (UL) assessment and maternal recall of the date of the last menstrual period (LMP) are two commonly used methods for determining gestational age. The information available in the registers varies between countries and over time. For example, in the United States, vital statistics information on gestational age derived from LMP was the only available measure until 1989, when it was replaced by a clinical estimate definition (though not reported from California). The US birth certification was revised in 2003, and the best obstetric estimate of gestational age was incorporated [6], including ultrasound estimation but excluding postnatal assessment [7]. These changes can affect regional and international comparisons as well as national trends. International studies suggest that the progressive use of early-ultrasound estimates could partially explain the stabilization and rise in preterm birth (accompanied by the fall in post-term births) in some countries in the world [8].

Although UL is usually considered to be the gold standard for gestational age estimation [7], it is known that the accuracy of the estimation of gestational age is strongly dependent on when the ultrasound is performed, which in turn is conditioned on when the mother has her first prenatal care visit [9]. For this reason, countries such as Sweden use the 'best estimate method' for clinical and public health surveillance, although studies are needed to evaluate the impact of this strategy [10]. Reliable information on gestational age is important in general but particularly when comparing outcomes between foreign- and native-born women, as any systematic difference in the estimation of gestational age between these groups could induce a biased association with no biological basis [11]. Furthermore, if the factors proposed to explain health differences in preterm birth between foreign and native mothers (e.g., socioeconomic variables) are also associated with the misclassification of the outcome, it would be unclear to what extent adjusting for such variables might explain health differences or reduce measurement error.

This study aims to examine whether disparities in gestational age outcomes between migrants and natives are contingent on the measure used to estimate gestational age and, if so, to identify which birth and maternal factors are associated with the discrepancy between LMP and UL. This study will contribute to a better understanding of the consequences of using different methods of estimation of gestational age for preterm and post-term surveillance among immigrants.

## Materials and methods

### Data

In this population-based register study, we used the Swedish Medical Birth Register (MBR), which contains maternal and child information on almost all births in Sweden [12]. Although the MBR started in 1973, we had access to data for the period 1992–2012. The MBR was linked to the Multi-Generational Register [13] to identify adopted mothers, and the Longitudinal Integration Database for Health Insurance and Labour Market Studies (LISA) [14] to obtain socio-economic indicators via an exclusive identification number [15].

## Study population

From a total of 2,239,806 observations, we excluded: 64,466 multiple births; 55,569 births with major congenital malformations; 1,115 adoptee women; and 55,788 women from other countries represented by less than 2,000 births. To ensure that we compared the same individuals throughout the study, we also excluded 723,253 observations with missing information in any of the following covariates: birthweight, maternal age, height, body mass index (BMI), smoking, country of birth, income, marital status and newborn's sex. Thereafter we excluded observations with biologically implausible combinations of birthweight and gestational age (n = 24,696) [16]. All subjects in the final sample (n = 1,317,265) had complete information on all gestational age measures available (Fig 1).

We also performed sensitivity analyses in a sub-sample of 765,030 uncomplicated pregnancies with spontaneous onset of delivery, in order to improve the accuracy of the ultrasound-based measures. Given that the estimation of gestational age using ultrasound depends on the size of the fetus, the accuracy of the measure is sensitive to factors that could influence fetal growth variations [17]. Hence, gestational age is systematically underestimated in smaller fetuses compared to normal-sized fetuses, while the opposite is true for those with rapid fetal growth [18–20]. Prior studies have indicated that both fixed (newborn's sex, maternal height) and modifiable (smoking, obesity) factors influence gestational age estimation [21, 22], and are thus implicated in the misclassification of preterm and post-term births [21]. For this reason, we conducted a sensitivity analysis in a sample of uncomplicated mothers with the aim of removing the potential effects of factors known to compromise the accuracy of the ultrasound-based estimates of gestational age. We defined uncomplicated pregnancies following the criteria used by INTERGROWTH-21st [23]. We excluded 765,030 observations corresponding to women with one or more of these characteristics: being younger than 18 or older than 34 at childbirth, with a pre-pregnancy BMI below 18.5 or above 30kg/m$^2$, a height below 153cm, smokers in early pregnancy who suffered from diabetes and hypertension before or during pregnancy and whose pregnancies ended in a cesarean section (Fig 1).

## Outcome variables

Two variables of gestational age were compared in this study: 1) date of the mother's last menstrual period (LMP-based); 2) expected date of birth based on ultrasound assessment (UL-based), which has replaced LMP dating for clinical decision making since 1990 in Sweden. Each of the available gestational age variables was divided into: 1) preterm (<37 weeks), term (37–41 weeks) and post-term (≥42 weeks); 2) very preterm (<32 weeks), moderately preterm (32–36 weeks), term (37–41) and post-term birth (≥42 weeks).

## Maternal birthplace

Migrants were defined according to their country of birth and grouped into seven geographical regions to warrant sufficient statistical power: Nordic countries, Europe & United States, Eastern Europe & Russia, Middle East, Africa, Asia and Latin America. The largest countries for each region were also selected (>2,000 observations) for specific country of birth analyses, which were conducted to assess heterogeneity within regional groupings.

## Covariates

Available covariates were: year of birth (categorized as: 1992–1995, 1996–1999, 2000–2004, 2005–2008 and 2009–2012); newborn's sex, parity (primiparous and multiparous); maternal height (linear and quadratic); maternal age (24 years or younger, 25–29, 30–34 and 35 years of

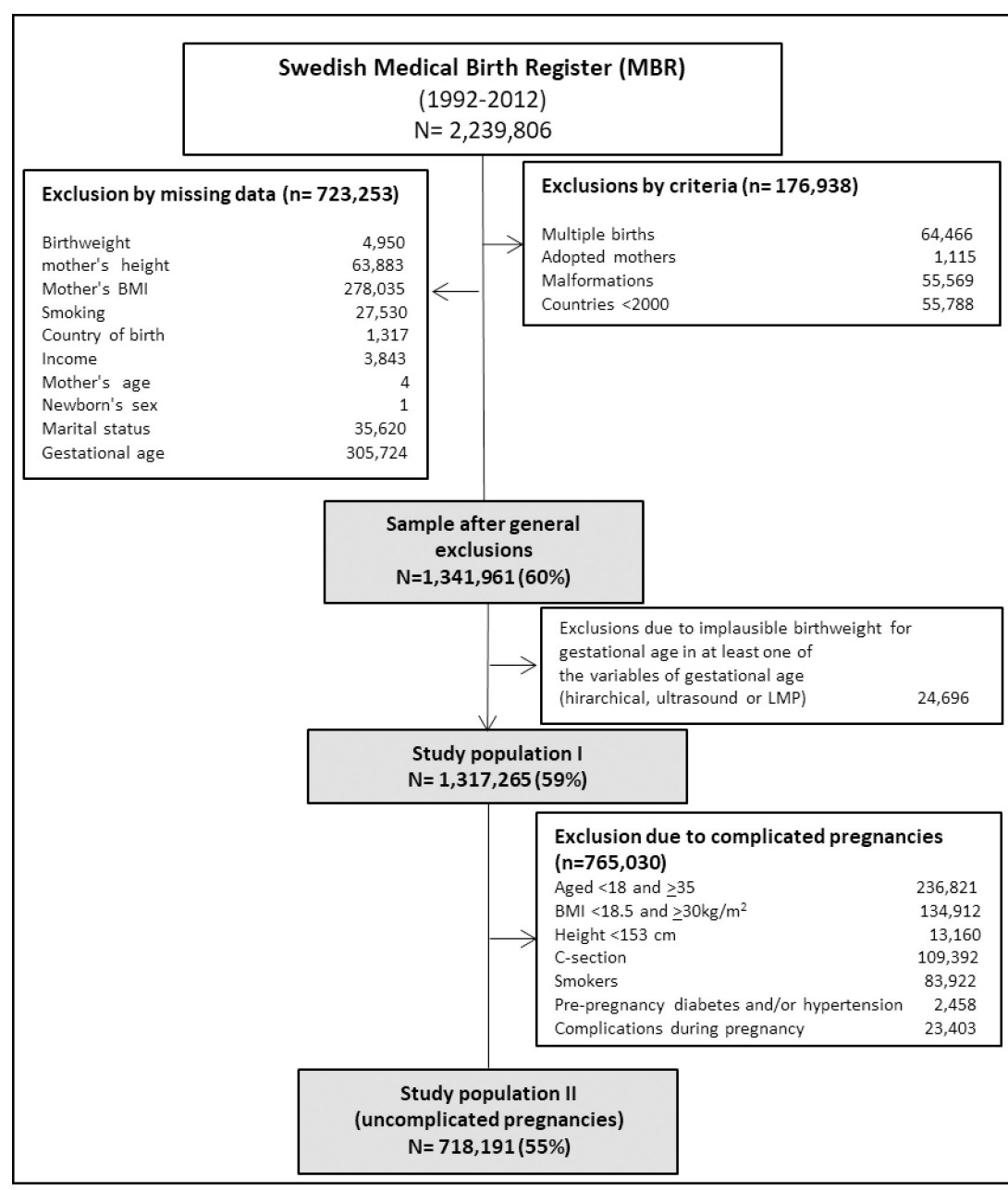

**Fig 1. Selection flow and study population.**

age and above); maternal BMI in early pregnancy, categorized as normal weight (18.50–24.99 kg /m2), underweight (<18.50 kg/m2), overweight (25.00–29.99 kg/m2), and obese (>30.00 kg/m2); maternal self-reported smoking in early pregnancy, between gestational weeks 10 and 12 (non-smokers, 1 to 9 cigarettes/day, 10 or more cigarettes/day); mother's educational attainment (low, middle and high); household disposable income (including salary, parental leave and other benefits, reported in quintiles); and maternal family situation (cohabiting with the father, single or other situation). All socio-demographic variables were considered for the year before birth.

## Statistical analysis

We first computed preterm and post-term birth by UL and LMP and estimated p-values using McNemar test for paired data to compare preterm and post-term rates between methods of gestational age estimation (UL-LMP) by mother's birthplace. Additionally, we estimated the Kappa coefficient (and their 95% Confidence Intervals, 95% CI) to assess the overall level of agreement between LMP and ultrasound categorized into preterm, term and post-term. The level of agreement in the Kappa coefficient is interpreted as follows: 0.20 (poor), 0.21–0.40 (fair), 0.41–0.60 (moderate), 0.61–0.80 (good) and 0.81–1.0 (very good) [24].

Second, we assessed the effect of using different measures of gestational age on disparities in the gestational age outcomes between the foreign-and Swedish-born. For this purpose, we used multinomial logistic regressions to derive odds ratios (OR) and 95% CI. We ran a total of eight models. First, we performed a set of two models (one based on ultrasound and the other on LMP), considering three categories of gestational age (preterm, term [reference outcome] and post-term) and using mother's birthplace as foreign vs Swedish-born. Second, we performed another set of two models with the same description but now disaggregating the preterm category into very preterm and moderately preterm. Third, we replicated the above strategy running the additional four models considering mother's birthplace into specific regions to see whether the main results were driven by a specific group of immigrants. The results are presented in forest plots and the estimates are included in tables as supporting material (S1 Table), together with additional models by mother's country of birth (S2 and S3 Tables). Additionally, we performed sensitivity analyses to assess the extent to which our results are affected by exclusions due to missing values in several covariates (S4 Table), and we also replicated the analyses in a sample of uncomplicated pregnancies (S5 Table), to see whether the results were consistent.

Finally, we used binary logistic regression models to identify variables associated with the discrepancy in the gestational age categories between UL and LMP. Discrepancy is defined as those observations placed in different gestational age categories defined as very preterm, moderately preterm, term and post-term (Table 1). Models were run separately for foreign and Swedish-born women. To evaluate the impact of the associated variables in the comparison between foreign and Swedish-born, we present unadjusted and adjusted estimates from multinomial logistic models considering gestational age outcomes as: very preterm, moderately preterm, term [reference outcome] and post-term.

All models were clustered by maternal identification number in order to obtain more accurate standard errors due to the existence of siblings in the dataset. All analyses were performed using Stata Statistical Software: Release 13, software (StatCorp, LP, College Station, Texas).

This study was approved by the Central Ethical Review Board of Stockholm in 2013 (decision no 2013/1058-32)

# Results

Discrepancies in gestational age outcomes by method of estimation (UL and LMP) are observed in foreign and Swedish-born women. Relative to UL, LMP classifies 16.7% of records in different categories. LMP has a wider gestational age distribution than UL, resulting in 17.8% and 20.9% of discrepant records classified as shorter gestations and 82.2% and 78.3% as longer gestation, among Swedish and foreign-born, respectively (Table 1).

## Differences in prevalence

Differences in preterm and post-term birth rates by gestational age estimation methods are shown for all women (Table 2). Preterm rates vary between methods among mothers born in

**Table 1. Discrepancies in gestational age outcomes between methods of estimation of gestational age (Ultrasound [UL], Last Menstrual Period [LMP]) by mother's birthplace.**

| UL | LMP | Swedish-born | Foreign-born |
|---|---|---|---|
| **Agreement (ref)** | | **914,398** | **182,446** |
| Term | Term | 770,432 | 157,218 |
| Very preterm | Very preterm | 4,983 | 1,184 |
| Moderately preterm | Moderately preterm | 20,257 | 3,882 |
| Post-term | Post-term | 118,726 | 20,162 |
| **Disagreement** | | **183,601 (100%)** | **36,820 (100%)** |
| **Underestimation** | | **32,514 (17.8%)** | **7,707 (20.9%)** |
| Moderately preterm | Very preterm | 0.21% | 0.24% |
| Term | Very preterm | <0.01% | 6 (0.02) |
| Term | Moderately preterm | 2.37% | 4.39% |
| Post-term | Very preterm | ND | ND |
| Post-term | Moderately preterm | 0.02% | 0.04% |
| Post-term | Term | 15.31% | 16.48% |
| **Overestimation** | | **149702 (82.2%)** | **28,823 (78.3%)** |
| Very preterm | Moderately preterm | 0.54% | 0.55% |
| Very preterm | Term | 0.01% | 0.01% |
| Very preterm | Post-term | <0.01% | ND |
| Moderately preterm | Term | 4.48% | 4.32% |
| Moderately preterm | Post-term | 0.06% | 0.07% |
| Term | Post-term | 76.98% | 73.89% |
| **Total No of births** | | **1,097,999** | **219,266** |

ND = No Data.

Sweden (UL: 3.19%; LMP: 2.83%, p-value <0.001), Africa (UL: 2.46%; LMP: 3.22%, p-value <0.001) and Latin America (UL: 3.64%; LMP: 3.29, p-value <0.01). Larger differences were observed when comparing post-term births across methods of estimation of gestational age

**Table 2. Preterm and post-term rates by method of estimation of gestational age and region of origin.**

| | | Preterm rates | | | Post-term rates | | | Level of agreement | |
|---|---|---|---|---|---|---|---|---|---|
| | | UL | LMP | | UL | LMP | | Agreement | |
| | Live births | % | % | P value | % | % | P value | Kappa Coef. | 95%CI |
| Swedish-born | 1,097,999 | 3.19 | 2.83 | <0.001 | 13.38 | 23.70 | <0.001 | 0.532 | [0.530.0.533] |
| Foreign-born | 219,266 | 3.18 | 3.19 | 0.6868 | 11.97 | 21.61 | <0.001 | 0.530 | [0.499,0.506] |
| **By region** | | | | | | | | | |
| Nordic | 26,735 | 3.03 | 2.89 | 0.0463 | 12.93 | 22.19 | <0.001 | 0.536 | [0.529,0.542] |
| Western Europe & USA | 8,619 | 2.38 | 2.20 | 0.1193 | 12.86 | 21.95 | <0.001 | 0.540 | [0.520,0.553] |
| Eastern Europe & Russia | 48,331 | 3.19 | 3.14 | 0.3792 | 12.47 | 21.68 | <0.001 | 0.513 | [0.501,0.520] |
| Middle East | 72,809 | 2.97 | 2.94 | 0.4297 | 10.02 | 20.70 | <0.001 | 0.480 | [0.471,0.487] |
| Africa | 23,061 | 2.46 | 3.22 | <0.001 | 23.04 | 28.94 | <0.001 | 0.509 | [0.502,0.519] |
| Asia | 28,845 | 4.47 | 4.42 | 0.6062 | 7.25 | 17.94 | <0.001 | 0.472 | [0.464,0.841] |
| Latin America | 10,866 | 3.64 | 3.29 | 0.0042 | 8.80 | 19.99 | <0.001 | 0.477 | [0.460,0.791] |

UL = Ultrasound, LMP = Last Menstrual Period.

P-values indicate comparisons for preterm and post-term rates between UL and LMP within each mother's origin.

The level of agreement in the Kappa coefficient is interpreted as follows: 0.20 (poor), 0.21–0.40 (fair), 0.41–0.60 (moderate), 0.61–0.80 (good) and 0.81–1.0 (very good).

(~10 points estimate) in all groups (p-value <0.001). The overall level of agreement between LMP and UL is moderately good regardless of mother's region of birthplace (Kappa coefficient ranging between 0.47 and 0.54 for Asia and Western Europe & USA, respectively) (Table 2).

## Gestational age disparities by mother's birthplace and method of gestational age estimation

The comparison of gestational age outcomes (preterm and post-term) between the offspring of foreign and Swedish-born mothers is contingent on the method used to estimate gestational age. The results from ultrasound showed that overall, compared to natives, foreign-born mothers had equal odds of having preterm births (OR: 0.98, 95% CI 0.96,1.01), lower odds of having moderately preterm and post-term births (respectively, OR: 0.95, 95%CI 0.92,0.98; OR:0.88, 95%CI 0.87,0.89), and higher odds of having very preterm births (OR: 1.13, 95%CI 1.07;1.20). However, the results derived from the LMP-based estimation led to consistently higher odds across all gestational age categories compared to term, except for post-term (Fig 2 and S1 Table). These inconsistencies were primarily driven by particular regional groups in each gestational age category (Fig 3 and S2 and S3 Tables). For example, in relation to preterm birth, the differences were mainly driven by mothers from Africa (OR$_{UL}$: 0.87, 95%CI 0.80,0.95 and OR$_{LMP}$: 1.23, 95%CI 1.15,1.34) and, to a lesser extent, mothers from Eastern Europe & Russia (OR$_{UL}$: 0.99, 95%CI 0.94, 1.05 and OR$_{LMP}$: 1.09, 95%CI: 1.03, 1.15), the Middle East (OR$_{UL}$: 0.90, 95%CI 0.86, 0.94 and OR$_{LMP}$: 1.00, 95%CI 0.96,1.05) and Latin America (OR$_{UL}$: 1.09, 95%CI 0.98, 1.21 and OR$_{LMP}$: 1.12, 95%CI 1.00,1.25).

The results obtained for all foreign-born mothers were consistent with results obtained from a sample of uncomplicated pregnancies (S4 Table), and also when observations with missing in other covariates were included (S5 Table).

## Variables associated with the differences between estimation methods of gestational age

The discrepancy in gestational age outcomes between methods of estimation of gestational age were, overall, similar for migrants and Swedes, with some exceptions: year of birth was associated with the difference between methods only among Swedes, showing consistently better agreement after 1995 compared to 1991–1995. Multiparous mothers (as opposed to primiparous) were associated with a lower odd (OR: 0.95, 95%CI 0.94,0.96) for discrepancy if they were Swedish-born but with higher odds if they were foreign-born (OR: 1.04, 95% CI 1.00,1.08). Similarly, while no differences were found in relation to marital status among Swedish mothers, a higher odd was found among non-married or cohabiting foreign-mothers (OR: 1.11, 95%CI 1.06,1.15) compared to married ones. Conversely, as compared to non-smoking, light smoking (9 cigarettes or less per day) during pregnancy was associated with a higher odd of discrepancy between methods only among Swedish mothers (OR: 1.07, 95%CI 1.05,1.09). Father's origin (being a foreign vs Swedish-born) was associated with a lower odd of misclassification only among Swedes (OR: 0.96 95%CI 0.94,0.98) (Table 3).

Models adjusted for the variables associated with the discrepancy between ultrasound and LMP generally contribute to reducing differences in gestational age outcomes between foreign and Swedish-born mothers. The reduction affects gestational age outcomes estimated with ultrasound and LMP (Fig 4). The exceptions are for ultrasound-estimated preterm births and for LMP-estimated post-term.

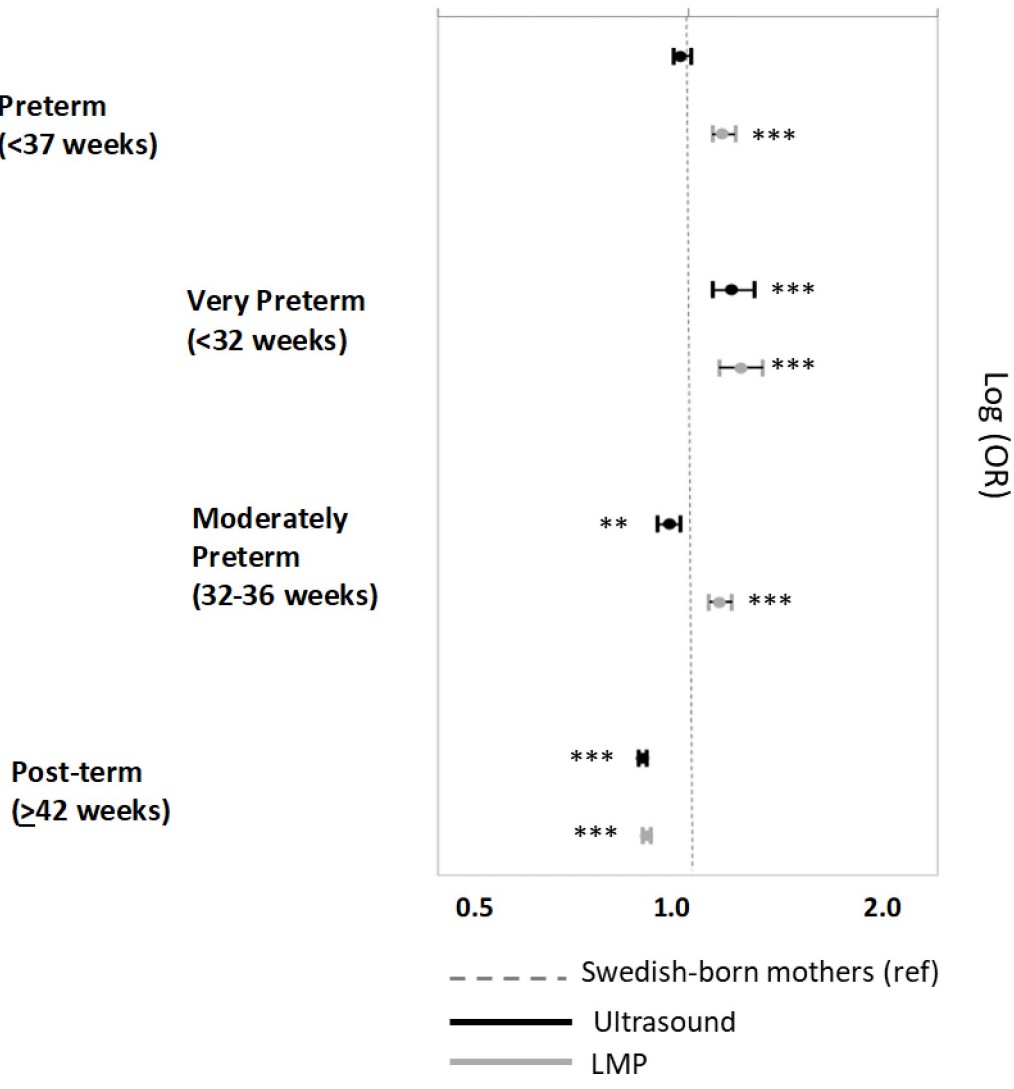

**Fig 2. Gestational age outcomes according to ultrasound and LMP estimates for the offspring of foreign-born women compared to Swedish-born women.** The stars indicate the level of significance comparing foreign-born and Swedish-born women for each model * $p < 0.05$, ** $p < 0.01$, *** $p < 0.001$.

## Discussion

### Summary of findings

Our results indicate that disparities in gestational age outcomes by mother's birthplace depend on the method used to estimate gestational age. While a health advantage is observed in almost all categories of gestational age when using ultrasound-based estimates (except for very pre-term), foreign-born mothers exhibit worse outcomes when LMP dating is used. However, these differences do not affect all migrant groups equally; the largest differences are found in mothers coming from Africa and, to a lesser extent, those from Eastern Europe and Russia, and the Middle East. Year of birth, multiplicity, marital status, father's origin and smoking were differentially associated with the disagreement between ultrasound and LMP estimates among foreign- and Swedish-born mothers. This finding suggests that controlling for these

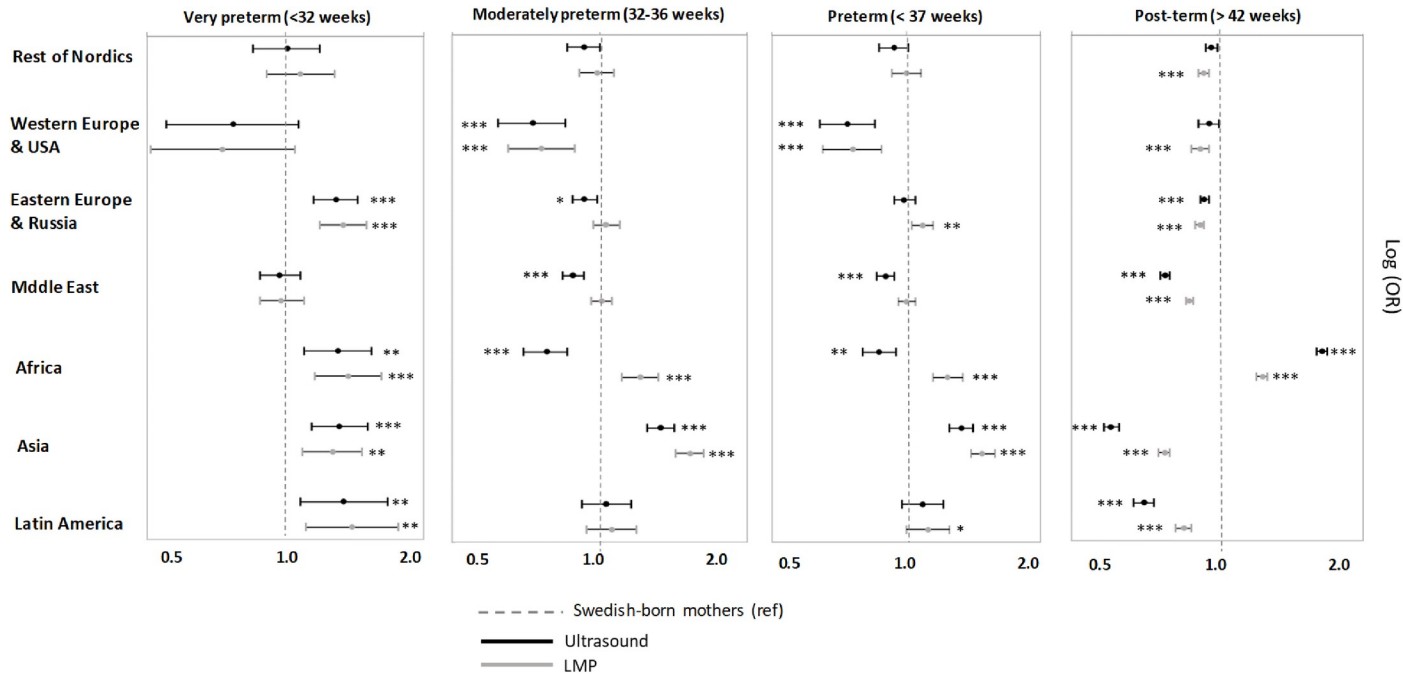

**Fig 3. Gestational age outcomes according to LMP and ultrasound estimates for the offspring of foreign-born women (by region of origin) compared to Swedish-born women.** The stars indicate the level of significance comparing foreign-born and Swedish-born women for each model * p<0.05, ** p<0.01, ***p<0.001.

variables (to reduce confounding or to evaluate mediation) could be problematic because they might reduce measurement error in the outcome rather than explain health differences between immigrants and natives.

### Links with previous research

In our study immigrants show lower odds of preterm births using ultrasound-based estimates. Although this result is in line with international evidence of a health advantage among immigrants compared to natives (healthy migrant paradox) [25], this finding is to some extent surprising in the Swedish context for at least two reasons. First, Sweden has some of the lowest rates of preterm birth in Europe (5.9% in 2009, including multiple births) [26] and the immigrant health advantage is observed among women from countries in conflict, who are likely to be refugees and who may have been exposed to adverse conditions before, during, and after migration. This advantage demonstrated among refugees contrasts with prior evidence from other national contexts [27]. Second, a recent study using the same population as our current study has shown that foreign-born mothers in Sweden have a higher risk of low birthweight deliveries [28], suggesting that migrants simultaneously have lower odds of preterm birth. These combined findings are puzzling since birthweight and gestational age are strongly correlated measures (i.e., fetal growth over time). Unlike gestational age, birthweight does not have differential measurement error by mother's origin (i.e., all births are weighed in the same way). As such, although ultrasound has been regarded as the most accurate method to estimate gestational age, results on preterm birth based on LMP estimations are surprisingly more consistent with the results that show higher risks of low birthweight.

Our findings are in line with previous studies which have shown that the LMP estimate is consistently associated with higher post-term birth, while mixed results have been seen in

**Table 3. Odds of the association between discrepant ultrasound-LMP information affecting the determination of gestational age outcomes and birth characteristics among foreign-born and Swedish-born mothers.**

| | Foreign-born | | | Swedish-born | | |
|---|---|---|---|---|---|---|
| | OR | 95%CI | p value | OR | 95%CI | p value |
| **Year of birth** | | | | | | |
| 1991–1995 (ref) | 1 | | | 1 | | |
| 1996–1999 | 0.99 | [0.95,1.04] | 0.742 | 0.98 | [0.96,1.00] | 0.016 |
| 2000–2004 | 0.98 | [0.94,1.02] | 0.372 | 0.95 | [0.94,0.97] | <0.001 |
| 2005–2008 | 0.99 | [0.95,1.03] | 0.601 | 0.95 | [0.93,0.97] | <0.001 |
| 2009–2012 | 0.98 | [0.94,1.02] | 0.254 | 0.92 | [0.90,0.94] | <0.001 |
| **Newborn's sex** | | | | | | |
| Male (ref) | 1 | | | 1 | | |
| Female | 1.11 | [1.08,1.13] | <0.001 | 1.12 | [1.11,1.13] | <0.001 |
| **Parity** | | | | | | |
| Primiparous (ref) | 1 | | | 1 | | |
| Multiparous | 1.04 | [1.00,1.08] | 0.007 | 0.95 | [0.94,0.96] | <0.001 |
| **Maternal Height (linear)** | 1.04 | [1.01,1.06] | 0.043 | 0.99 | [0.95,1.03] | 0.655 |
| **Mother's BMI** | | | | | | |
| Normal weight (ref) | 1 | | | 1 | | |
| Underweight | 0.95 | [0.90,1.01] | 0.094 | 0.98 | [0.95,1.01] | 0.214 |
| Overweight | 1.13 | [1.10,1.16] | <0.001 | 1.08 | [1.07,1.09] | <0.001 |
| Obese | 1.33 | [1.28,1.38] | <0.001 | 1.26 | [1.24,1.28] | <0.001 |
| **Maternal age** | | | | | | |
| 25–29 (ref) | 1 | | | 1 | | |
| < = 24 | 1.12 | [1.08,1.15] | <0.001 | 1.19 | [1.17,1.20] | <0.001 |
| 30–34 | 0.88 | [0.85,0.90] | <0.001 | 0.87 | [0.86,0.88] | <0.001 |
| > = 35 | 0.75 | [0.73,0.78] | <0.001 | 0.74 | [0.73,0.76] | <0.001 |
| **Mother's education attaintment** | | | | | | |
| High (ref) | 1 | | | 1 | | |
| Low | 1.19 | [1.15,1.23] | <0.001 | 1.11 | [1.09,1.13] | <0.001 |
| Middle | 1.09 | [1.06,1.13] | <0.001 | 1.04 | [1.03,1.05] | <0.001 |
| **Household disposable income** | | | | | | |
| Botton (ref) | 1 | | | 1 | | |
| 2 | 0.99 | [0.96,1.02] | 0.351 | 0.96 | [0.94,0.98] | <0.001 |
| 3 | 0.95 | [0.92,0.99] | 0.014 | 0.92 | [0.91,0.94] | <0.001 |
| 4 | 0.96 | [0.92,1.01] | 0.103 | 0.90 | [0.88,0.91] | <0.001 |
| Top | 0.94 | [0.90,0.98] | 0.009 | 0.89 | [0.87,0.90] | <0.001 |
| **Marital status** | | | | | | |
| Married/cohabiting | 1 | | | 1 | | |
| Other | 1.11 | [1.06,1.15] | <0.001 | 0.99 | [0.97,1.02] | 0.262 |
| **Father's origin** | | | | | | |
| Swedish-born (ref) | 1 | | | 1 | | |
| Foreign-born | 1.00 | [0.97,1.03] | 0.988 | 0.96 | [0.94,0.98] | <0.001 |
| **Smoking during pregnancy** | | | | | | |
| No (ref) | 1 | | | 1 | | |
| 1 to 9 cig/per day | 1.01 | [0.96,1.05] | 0.710 | 1.07 | [1.05,1.09] | <0.001 |
| 10 + cig/per day | 0.96 | [0.89,1.03] | 0.235 | 0.98 | [0.96,1.01] | 0.262 |

OR = Odd Ratios; CI = Confidence Intervals.

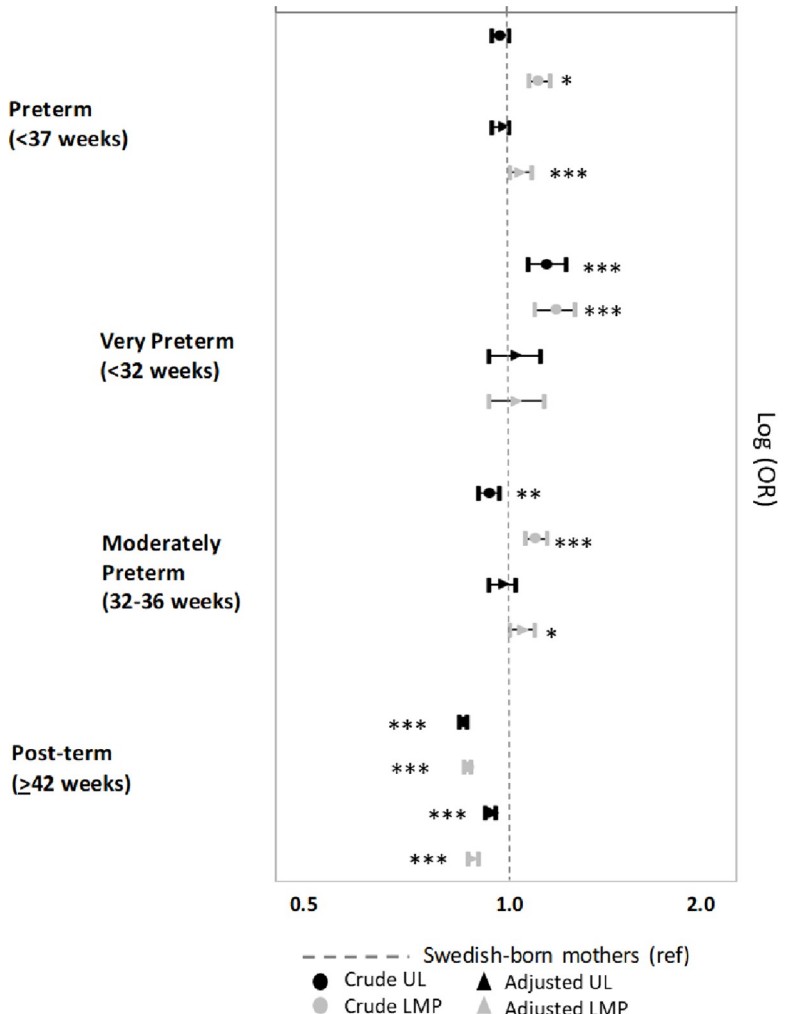

**Fig 4. Gestational age outcomes according to LMP and ultrasound estimates for the offspring of foreign-born women (by region of origin) compared to Swedish-born women.** Differences between unadjusted and adjusted models. Model adjusted for year of birth, newborn's sex, parity, maternal height, maternal BMI, maternal age, mother's educational attainment, household disposable income, marital status, father's origin, smoking during pregnancy. The stars indicate the level of significance comparing foreign-born and Swedish-born women for each model * p<0.05, ** p<0.01, ***p<0.001.

relation to preterm [7, 29, 30]. Prior studies have also demonstrated that the two gestational age methods differ according to maternal-level characteristics, such as obesity and smoking during pregnancy [21]. Our findings indicate that region and country of origin are also important factors to take into consideration.

Year of birth, multiplicity, marital status, father's origin and smoking were also differentially associated with the discrepancy between LMP and ultrasound depending on whether the mother is Swedish or a foreign-born. These variables do not explain the health advantage observed with ultrasound since consistent results were found in a sub-sample of uncomplicated pregnancies that exclude multiple births and smoker mothers. Furthermore, the results remain the same when single women and women with a Swedish partner are additionally excluded (results upon request). However, the association between these variables and the discrepancy between methods is of importance as this may lead to wrong conclusions when

trying to use this data to identify risk factors for a gestational age outcome [17, 21]. For example, we cannot exclude the possibility that an association between smoking and preterm (estimated with ultrasound) may reflect the existence of measurement error in the outcome (gestational ages of small fetuses are underestimated with ultrasound and smoking might lead to small fetuses) rather than a risk factor for preterm.

## Strengths and limitations

To the best of our knowledge, this is the first study to assess the implications of using different methods of gestational age estimation to study disparities in birth outcomes between immigrants and natives. This study is conducted using high quality and large population registers in a country context with a wide diversity of migrant origins. This provides enough statistical power to explore disparities by world regions and specific countries. Both ultrasound and LMP-based estimates of gestational age recorded in the MBR are measured in days, which allowed for a more precise evaluation of the discrepancy between the two dating methods than assessments based on weeks. The detailed information collected in the MBR also allows us to compare the results using a sub-sample of uncomplicated pregnancies. These analyses show that the differences observed between migrant and native mothers are not driven by uneven distributions of the observed risk factors, adverse environmental conditions, pathological process or obstetric interventions. However, other important variables are missing, including the date when the ultrasound was conducted or delays on the first antenatal visit, which might explain why larger differences are seen for some regions between estimation methods than others.

## Implications for future research and policy

Our findings may also be useful in the interpretation of national comparisons of disparities in preterm birth by migration status and other sociodemographic characteristics. For example, year of birth was associated with the difference between modes of estimation of gestational age, but only among Swedes. This suggests that the quality of the information could be responsible for artefactual differences in gestational age outcomes in some earlier studies conducted.

From our findings, we recommend focusing on very preterm in studies on migration, as this category not only is of greater clinical relevance but it shows higher risks among migrants. We also observed that the smallest discrepancy between methods of gestational age estimation is for very preterm. Furthermore, studies have shown that the accuracy of gestational age is higher among very preterm than among the broad category of preterm in validation studies that compared hospital and vital statistics records [31].

There is an ongoing debate on whether ultrasound should be the first choice for dating pregnancies and/or exclusively early ultrasound [8]. Although we cannot disentangle this concern, our findings are relevant for those contexts (such as Sweden) where the 'best estimate' method is used for clinical and public health surveillance. Our study shows that the results obtained with the 'best estimate' method could also be affected depending on the extent to which ultrasound or LMP is used across groups since they lead to different results.

Future research should strive to understand the measurement properties of the gestational age estimate being used. Although our sensitivity analysis using a sample of uncomplicated pregnancies strengthens the accuracy of the ultrasound-based estimate (since under optimal conditions fetal growth should be similar worldwide), it does not rule out the possibility that other factors may influence the accuracy of the ultrasound estimation. For example, future studies may examine the extent to which the discrepancy between UL and LMP observed in this study is associated to the fact that foreign-born expectant mothers tend to arrive late to

their first prenatal visit [32]. It is known that up to week 13 6/7 weeks of gestation, the ultrasound estimation of gestational age based on measurement of crown–rump length (CRL) has an error of ± 5–7 days while, after week 14, measurements with lower accuracy (e.g., fetal biparietal diameter, femur length, abdominal circumference or length of the humerus) have an estimated error of approximately 14 days between weeks 22 0/7 and 27 6/7 [33]. Prior perinatal research has demonstrated that discrepant information between UL and LMP *per se* is associated with adverse perinatal outcomes (such as infant mortality and morbidity) [34, 35], which might well be a consequence of poor early prenatal monitoring.

Future research on migrant disparities in preterm birth should exercise caution when interpreting the international literature. Given that preterm disparities between foreign and Swedish-born is contingent on the method used to estimate gestational age (lower OR of preterm using ultrasound but higher odds using LMP), our study suggests that the healthy migrant paradox [36–38] might to some extent depend on the method used to estimate gestational age, which varies between countries and within countries over time.

In this study we have focused on the implications of the UL-LMP discrepancy for gestational age surveillance in the context of migration and perinatal health research. However, the discrepancy observed between methods could also have clinical implications that are beyond the scope of this study but which could nevertheless influence public health surveillance of preterm outcomes (e.g. through selection bias). Gestational age is not only used to derive perinatal health outcomes (such as those examined in this study), but also to decide whether intrauterine death should be defined as stillbirth or fetal death, and to determine when a pregnancy reaches the viability threshold. This is necessary, for example, in order to determine the legality of any abortions that are performed [39]. Furthermore, gestational age is used to decide when induction of labor should start (generally after 41 weeks) [40].

## Conclusions

The direction and strength of disparities in gestational age outcomes are affected by the method of estimation used, as do some of the risk factors associated with the timing of birth outcomes. We recommend focusing on very preterm in studies on migration, since this is the gestational age outcome that shows consistently higher risks among migrants across methods of gestational age estimation.

## Supporting information

**S1 Table. Gestational age outcomes by method of estimation of gestational age by mother's birthplace (corresponding to Figs 2 and 3).**
(DOCX)

**S2 Table. Preterm according to LMP and ultrasound estimates by mother's country of birth.**
(DOCX)

**S3 Table. Post-term according to LMP and ultrasound estimates by mother's country of birth.**
(DOCX)

**S4 Table. Sensitivity analyses.** Gestational age outcomes according to LMP and ultrasound estimates in a subsample of 718,191 of uncomplicated pregnancies.
(DOCX)

**S5 Table. Sensitivity analyses.** Gestational age outcomes according to LMP and ultrasound estimates including 723,253 missing in different covariates.
(DOCX)

**S6 Table. Gestational age outcomes by method of estimation of gestational age by region of origin adjusted for maternal variables (corresponding to Fig 4).**
(DOCX)

## Author Contributions

**Conceptualization:** Sol P. Juárez, Marcelo L. Urquia, Eleonora Mussino, Can Liu, Anders Hjern.

**Data curation:** Sol P. Juárez, Marcelo L. Urquia.

**Formal analysis:** Sol P. Juárez, Marcelo L. Urquia.

**Funding acquisition:** Sol P. Juárez, Anders Hjern.

**Investigation:** Sol P. Juárez, Marcelo L. Urquia.

**Methodology:** Sol P. Juárez, Marcelo L. Urquia, Can Liu, Anders Hjern.

**Writing – original draft:** Sol P. Juárez, Marcelo L. Urquia.

**Writing – review & editing:** Sol P. Juárez, Marcelo L. Urquia, Eleonora Mussino, Can Liu, Yao Qiao, Anders Hjern.

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
