## [Decision Letter · Decision Letter 0]

12 Nov 2020

PONE-D-20-31406

Preterm disparities between migrants and native Swedes depend on the method used to estimate gestational age. A Swedish population-based register study

PLOS ONE

Dear Dr. Juárez,

Thank you for submitting your manuscript to PLOS ONE. After careful consideration, we feel that it has merit but does not fully meet PLOS ONE’s publication criteria as it currently stands. Therefore, we invite you to submit a revised version of the manuscript that addresses the points raised during the review process.

We look forward to receiving your revised manuscript.

Kind regards,

Kelli K Ryckman

Academic Editor

PLOS ONE

Journal Requirements:

Reviewers' comments:

Reviewer's Responses to Questions

**Comments to the Author**

1. Is the manuscript technically sound, and do the data support the conclusions?

Reviewer #1: Partly

Reviewer #2: Yes

2. Has the statistical analysis been performed appropriately and rigorously? 

Reviewer #1: Yes

Reviewer #2: Yes

3. Have the authors made all data underlying the findings in their manuscript fully available?

Reviewer #1: Yes

Reviewer #2: No

4. Is the manuscript presented in an intelligible fashion and written in standard English?

Reviewer #1: Yes

Reviewer #2: Yes

5. Review Comments to the Author

Reviewer #1: Preterm disparities between migrants and native Swedes depend on the method used to estimate gestational age. A Swedish population-based register study

Juarez et al use a Swedish population register to investigate the disparity in classifying births as preterm or post-term between menstrual period and ultrasound in foreign and Swedish-born woman.

The idea is novel: While it is well known that there is a discrepancy between those 2 methods, the clinical implications are not well studied.

I have a few major concerns/suggestions:

First, as the authors mention in the introduction of their paper, the accuracy of the US methods depends on when it is done during gestation and as a consequence depends on prenatal care. The authors do not present any data on prenatal care. Do the authors have any data on this? It is often collected in administrative databases. Could prenatal care be assessed as a mediator in the difference between the methods?

Table 2 would best be converted into a forest plot. It is very difficult to read. I am also worried about multiple comparison. Did the authors account for that? To that point, it would be nice to present the p-value in addition to the OR to see to what degree these results are sensitive to a Bonferroni (or similar) correction.

Table 3 is interesting but only shows half of the story. It shows that the discrepancy is higher for a lot of the socioeconomic variables, again this reviewer is wondering why there is a discrepancy, maybe it is all related to late initiation of prenatal care? If that is the case, then the discrepancy is not that important but rather the timing of initiation of prenatal care.

Generally, I think the manuscript could benefit from some more “user friendly” explanations of the finding. It took this reviewer a while to understand the data and the results. I think the manuscript is well written though and the analysis is interesting, it is just difficult to grasp and maybe an additional figure or example would help.

Reviewer #2: Thanks you for the opportunity to review this interesting manuscript based on a large Scandinavian population cohort.

Page 6, line 163 “Thereafter we excluded observations with biologically implausible combinations of birthweight and gestational age” please provide details and/or reference for the method by which you determined the parameters for biological impausibility for exclusion.

Although I understand why for clarify and simplicity it was tempting for the authors to use logistic regression to break GA into categories and use logistic regression, this results in a huge loss of information when in fact the authors had access to very granular GA estimates in days and could have looked at discrepancies in more detail rather that just as black and white yes/no agreement of clinical categories with problematic edge effects (ie 36 weeks and 6 days is preterm while 37 weeks and 0 days is term despite a discrepancy of one day--and this discrepancy would be treated equivalently to a discrepancy of 100 days by a dichotomous model)

Page 8, Line 231: We considered as ‘discrepant’ those observations placed in different gestational age categories, that is, preterm according to ultrasound-based, and term according to LMP-based estimates. Does this imply that the analysis presented in Table 3 only includes observations for which gestational category was discrepant? If so, please clarify this in the text, and also provide details of the sample size for this analysis. Or does this imply that discrepant vs. concordant was used at the dichotomous outcome for this analysis? This is not clear. This analytical strategy may also be flawed as it does not account for directionality (ie it treats misclassification from term to preterm the same as preterm to term—please clarify how the outcome was coded for this analysis, and what cohort was included.).

Page 9, line 253 and elsewhere, you have not provided a formal comparison to support your assertion that preterm rates were more consistent in foreign vs native mothers (granted the confidence intervals are very tight and non-overlapping, but for cases where the differences are not as clear, a formal difference test would be necessary to support these assertions.

Page 9 Line 255:

“With the exception of African mothers, preterm rates overlapped between methods of estimation in all groups”. This statement isn’t meaningful from a statistical standpoint. As above with respect to consistency, you haven’t provided a formal statistical comparison. Confidence intervals for two comparitors can overlap substantially and still be significantly different. If you want to make claims of similarity or differences please formalize your comparisons with statistical tests-but note that comparing

Page 11 Line 288-298

“Lower odds vs. higher odds” these are all relative so “lower” odds in one group merely means misclassification is higher in one group vs the other. Please present CIs for these in the text if presenting them. I find this analysis a little concerning/problematic as misclassification can be in either direction and has very asymmetric consequences.

Table 1, and elsewhere For all significance tests/p-values: Please do not use asterices to signify levels of significance. Provide exact p-values unless p-value is <0.0001.

Discussion/Conclusions: It’s unclear how the impact of the factors studied on misclassification by LMP vs. US when it doesn’t appear that directionality of the bias was considered. The results presented and discussed from Table 3 appear to be factors that were associated with discrepant measurements, ignoring the direction or severity of the discrepancy. This makes interpretation in a broader context difficult. If I am interpreting the construction of the analysis wrong, please make it more clear in the methods section and presentation of results.

6. PLOS authors have the option to publish the peer review history of their article (what does this mean?). If published, this will include your full peer review and any attached files.

Reviewer #1: No

Reviewer #2: **Yes: **Steven Hawken

---

## [Author Response · Author response to Decision Letter 0]

14 Jan 2021

Thank you very much for your constructive comments. Below is the point-by-point response to your concerns. 

Reviewer 1 #1: Preterm disparities between migrants and native Swedes depend on the method used to estimate gestational age. A Swedish population-based register study

Juarez et al use a Swedish population register to investigate the disparity in classifying births as preterm or post-term between menstrual period and ultrasound in foreign and Swedish-born woman.

The idea is novel: While it is well known that there is a discrepancy between those 2 methods, the clinical implications are not well studied.

I have a few major concerns/suggestions:

First, as the authors mention in the introduction of their paper, the accuracy of the US methods depends on when it is done during gestation and as a consequence depends on prenatal care. The authors do not present any data on prenatal care. Do the authors have any data on this? It is often collected in administrative databases. Could prenatal care be assessed as a mediator in the difference between the methods?

AUTHORS: Very unfortunately, information on timing for the prenatal care visits is not available in the Swedish Medical Birth Register. It is therefore impossible for us to determine the extent to which the discrepancy between methods of estimation of gestational age is due to a delay on the first prenatal visit. However, we have explicitly mentioned the need to evaluate this mechanism (see pages 22-23, lines 433-443) and also highlighted this lack of information as a limitation (see page 21, lines 403-406).

Reviewer 1 #2: Table 2 would best be converted into a forest plot. It is very difficult to read. I am also worried about multiple comparison. Did the authors account for that? To that point, it would be nice to present the p-value in addition to the OR to see to what degree these results are sensitive to a Bonferroni (or similar) correction.

AUTHORS: Thank you for your suggestion. We now include forest plots to present the main results and include the former table 2 (including p-values) in the appendix (for the relevance of future meta-analyses).

In the revised version we include p-values in all tables, as requested. However, we believe that multiple comparison is not a concern in our case for two main reasons. First, we use well-established categories in the literature of migration and health. Migration status (foreign vs natives), and categorizations by geographic proximity or countries of origin are very common in the literature. Membership to countries of origin is not a random process and we are not using a sample but the whole population. Therefore, it is very unlikely that our observed associations are due to chance. Second, we examine foreign-born mothers in a stepwise fashion trying to disentangle whether the overall results (from the foreign vs Swedish-born comparison) are driven by specific groups of immigrants. In relation to this last point, we would like to clarify that the main results come from eight independent multinomial models and some of them are quite parsimonious. We realized, however, that the description of the model strategy was not very clear in the previous version of our manuscript so we have now clarified this point in the revised version. See pages 8-9 lines, 192-205.

Reviewer 1 #3: Table 3 is interesting but only shows half of the story. It shows that the discrepancy is higher for a lot of the socioeconomic variables, again this reviewer is wondering why there is a discrepancy, maybe it is all related to late initiation of prenatal care? If that is the case, then the discrepancy is not that important but rather the timing of initiation of prenatal care.

AUTHORS: We would like to clarify that it is beyond the scope of this paper to explain the differences between methods of gestational age estimation. We identify the birth and maternal characteristics associated with the discrepancies with the purpose of further discussing the consequences of using different methods in the study of migration and perinatal health, not to explain the discrepancies between methods. Specifically, we want to examine whether variables commonly used as confounders or mediators in studies on preterm disparities by mother’s birthplace are also associated with the misclassification of the outcome. This is important as the inclusion of these variables in the model could reduce measurement error rather than explain health differences between groups. In the revised version we try to better explain the purpose of table 3. To that effect, we introduced the rationale in the introduction (see page 4, lines 99-103), included additional analyses showing how the adjustments for these maternal variables modify the differences between foreign-born and Swedish-born women (see figure 4; page 17, lines 316-320) and we interpreted the results in the discussion section (see page 19 lines 346-349).

Although we do not offer an explanation of the observed discrepancies, we believe that our findings could inspire further research in this direction. In this regard, we explicitly elaborate on how timing for prenatal care between groups could largely explain the observed discrepancy (see pages 22-23, lines 433-443).

Reviewer 1 #4: Generally, I think the manuscript could benefit from some more “user friendly” explanations of the finding. It took this reviewer a while to understand the data and the results. I think the manuscript is well written though and the analysis is interesting, it is just difficult to grasp and maybe an additional figure or example would help.

AUTHORS: In the revised version, we try to better guide the reader, especially through the method and result sections. We made our model strategy more explicit (see page 8-9), clarified the definition of some outcomes (see table 1), and we tried to be more upfront with our rationale throughout (see page 4, lines 99-103; page 9 lines, 207-214; page 10 lines 234-238; page 17, lines 317-320, figure 4, page 19, lines 346-349). All in all, we hope the final version makes for more pleasant reading. 

Reviewer #2: 

Reviewer 2 #1: Thanks you for the opportunity to review this interesting manuscript based on a large Scandinavian population cohort.

Page 6, line 163 “Thereafter we excluded observations with biologically implausible combinations of birthweight and gestational age” please provide details and/or reference for the method by which you determined the parameters for biological impausibility for exclusion.

AUTHORS: For the identification of implausible combinations of birthweight and gestational age we have used the thresholds based on fetal growth references published for Sweden. We have now included the corresponding reference. See reference #16 (page 5, line 128)

Källén B. A birth weight for gestational age standard based on data in the Swedish Medical Birth Registry, 1985-1989. European Journal of Epidemiology. 1995;11(5):601-6.

Reviewer 2 #2: Although I understand why for clarify and simplicity it was tempting for the authors to use logistic regression to break GA into categories and use logistic regression, this results in a huge loss of information when in fact the authors had access to very granular GA estimates in days and could have looked at discrepancies in more detail rather that just as black and white yes/no agreement of clinical categories with problematic edge effects (ie 36 weeks and 6 days is preterm while 37 weeks and 0 days is term despite a discrepancy of one day--and this discrepancy would be treated equivalently to a discrepancy of 100 days by a dichotomous model)

Page 8, Line 231: We considered as ‘discrepant’ those observations placed in different gestational age categories, that is, preterm according to ultrasound-based, and term according to LMP-based estimates. Does this imply that the analysis presented in Table 3 only includes observations for which gestational category was discrepant? If so, please clarify this in the text, and also provide details of the sample size for this analysis. Or does this imply that discrepant vs. concordant was used at the dichotomous outcome for this analysis? This is not clear. This analytical strategy may also be flawed as it does not account for directionality (ie it treats misclassification from term to preterm the same as preterm to term—please clarify how the outcome was coded for this analysis, and what cohort was included.).

AUTHORS: The analysis presented in table 3 shows the variables associated with the discrepancy between methods of estimation of gestational age (LMP-UL). The outcome is coded as agreement between ultrasound and LMP (reference) vs disagreement. We only considered ‘discrepancies’ when the differences between methods of estimation led to substantial changes in well-defined health outcomes. In the previous version, we used only three categories (preterm, term and post-term) to determine discrepancies between methods. However, considering the reviewer’s comment, in the revised version, we also considered discrepancies using more categories (very preterm, moderately preterm, preterm, term and postterm), though the results were identical. To help clarify how the outcome was coded, we included a table with all the possible combinations including sample size and indicating the direction of the discrepancies (see table 1), in addition to expanding the description in the text (see page: 9 lines 207-214). 

The decision to examine discrepancies between methods (UL-LMP), using categories of gestational age (rather than days or weeks) while ‘ignoring’ the directionality of the discrepancies, is motivated by the purpose of this analysis. The aim of this analysis is to examine whether variables commonly used as confounders or mediators in studies on preterm disparities by mother’s birthplace are also associated with the misclassification of the outcome. As such, we believe that it makes more sense to define the discrepancies in alignment with how we classify the outcomes in the main analysis. The results suggested that both the exposure and the common explanatory factors (i.e. confounders and mediators) are associated with outcome misclassification. The misclassification of outcome is not trivial as the direction of the overall association can flip (as shown in forest plots). We understand that in the previous version of the manuscript the purpose of this secondary analysis was not clear enough. For this reason, in the revised version, we implemented a number of changes throughout: we introduced the aim (page 4, lines 99-103), included additional analyses to evaluate the impact of these variables in the comparison of gestational age outcomes between foreign and Swedish-born women (see figure 4 and page 17, lines 316-320), and interpreted the findings in the discussion section (page 19, lines 346-349). 

Reviewer 2 #3: The analysis were designed in order to be aligned with the main results. As such, we considered as ‘discrepant’ observations that were placed in different categories of gestational age (here as preterm vs term) between different methods of estimation. As such, 

Page 9, line 253 and elsewhere, you have not provided a formal comparison to support your assertion that preterm rates were more consistent in foreign vs native mothers (granted the confidence intervals are very tight and non-overlapping, but for cases where the differences are not as clear, a formal difference test would be necessary to support these assertions.

Page 9 Line 255:

“With the exception of African mothers, preterm rates overlapped between methods of estimation in all groups”. This statement isn’t meaningful from a statistical standpoint. As above with respect to consistency, you haven’t provided a formal statistical comparison. Confidence intervals for two comparitors can overlap substantially and still be significantly different. If you want to make claims of similarity or differences please formalize your comparisons with statistical tests-but note that comparing 

AUTHORS: Thanks for pointing at this imprecise statement. We have rephrased it in the revised version. In order to formally compare preterm and postterm birth between ultrasound and LMP within origins, we implemented two changes in table 2. First, we included a column with p-values using McNemar test for paired data. Second, we included a column with a measure of overall agreement (Kappa coefficient) between ultrasound and LMP. See page 8, lines 182-187 in methods; page 11, lines, 244-251 in the results.

Although the reviewer’s comment refers to table 2, we would like to take the opportunity to clarify why we did not calculate p-values to formally compare the estimates derived from different multinomial models (using UL and LMP, see forest plots). In this case we did not consider it necessary to establish a formal comparison since we want to illustrate how, in practice, we can infer different conclusions from the comparison between immigrants and natives depending on whether we estimate gestational age outcomes from UL and LMP. Please, note that in most cases researchers do not have access to both UL and LMP to see how these results differ. This is the reason why the column “consistent” in the tables presented in the appendix is not supported by a formal test. 

Reviewer 2 #4: Page 11 Line 288-298

“Lower odds vs. higher odds” these are all relative so “lower” odds in one group merely means misclassification is higher in one group vs the other. Please present CIs for these in the text if presenting them. I find this analysis a little concerning/problematic as misclassification can be in either direction and has very asymmetric consequences.

AUTHORS: In the revised version, we included confidence intervals when reporting the findings in the text and also clarified the comparison category (see page 13, lines 262-276). In a previous answer (see the response to your question #2) we hope we justified the decision not to include the directionality in the main analysis. However, we have included detailed information for the reader to see the most common type of misclassification (see table 1) and directionality (overestimation). See page 10, lines 234-238.

Reviewer 2 #4: Table 1, and elsewhere For all significance tests/p-values: Please do not use asterices to signify levels of significance. Provide exact p-values unless p-value is <0.0001.

AUTHORS: We have now added the corresponding p-values with four decimals in all tables. 

Reviewer 2 #5: Discussion/Conclusions: It’s unclear how the impact of the factors studied on misclassification by LMP vs. US when it doesn’t appear that directionality of the bias was considered. The results presented and discussed from Table 3 appear to be factors that were associated with discrepant measurements, ignoring the direction or severity of the discrepancy. This makes interpretation in a broader context difficult. If I am interpreting the construction of the analysis wrong, please make it more clear in the methods section and presentation of results.

AUTHORS: please see our answer to your question #2 above.

---

## [Decision Letter · Decision Letter 1]

2 Feb 2021

Preterm disparities between foreign and Swedish born mothers depend on the method used to estimate gestational age. A Swedish population-based register study

PONE-D-20-31406R1

Dear Dr. Juárez,

We’re pleased to inform you that your manuscript has been judged scientifically suitable for publication and will be formally accepted for publication once it meets all outstanding technical requirements.

Kind regards,

Kelli K Ryckman

Academic Editor

PLOS ONE

Additional Editor Comments (optional):

Reviewers' comments:

Reviewer's Responses to Questions

**Comments to the Author**

1. If the authors have adequately addressed your comments raised in a previous round of review and you feel that this manuscript is now acceptable for publication, you may indicate that here to bypass the “Comments to the Author” section, enter your conflict of interest statement in the “Confidential to Editor” section, and submit your "Accept" recommendation.

Reviewer #2: All comments have been addressed

2. Is the manuscript technically sound, and do the data support the conclusions?

Reviewer #2: Yes

3. Has the statistical analysis been performed appropriately and rigorously? 

Reviewer #2: Yes

4. Have the authors made all data underlying the findings in their manuscript fully available?

Reviewer #2: Yes

5. Is the manuscript presented in an intelligible fashion and written in standard English?

Reviewer #2: Yes

6. Review Comments to the Author

Reviewer #2: Thank you to the authors for their revised manuscript and thoughtful response to reviewer comments, I am satisfied that my concerns have been addressed in the revised version.

7. PLOS authors have the option to publish the peer review history of their article (what does this mean?). If published, this will include your full peer review and any attached files.

Reviewer #2: **Yes: **Steven Hawken

---

## [Editor Report · Acceptance letter]

11 Feb 2021

PONE-D-20-31406R1 

Preterm disparities between foreign and Swedish born mothers depend on the method used to estimate gestational age. A Swedish population-based register study 

Dear Dr. Juárez:

I'm pleased to inform you that your manuscript has been deemed suitable for publication in PLOS ONE. Congratulations! Your manuscript is now with our production department. 

Kind regards, 

on behalf of

Dr. Kelli K Ryckman 

Academic Editor

PLOS ONE